# Legume Protein Extracts: The Relevance of Physical Processing in the Context of Structural, Techno-Functional and Nutritional Aspects of Food Development

**Chaima Neji [1], Jyoti Semwal [1,2], Mohammad Hassan Kamani [3], Endre Máthé [1] and Péter Sipos [1,\*]**

1   Faculty of Agricultural and Food Sciences and Environmental Management, Institute of Nutrition, University of Debrecen, Böszörményi út 138, 4032 Debrecen, Hungary
2   Department of Grain Science and Technology, CSIR-Central Food Technological Research Institute, Mysore 570020, Karnataka, India
3   Food Chemistry and Technology Department, Teagasc Food Research Centre, Moorepark, P61 C996 Fermoy, County Cork, Ireland
\*   Correspondence: siposp@agr.unideb.hu

**Abstract:** Legumes are sustainable protein-rich crops with numerous industrial food applications, which give them the potential of a functional food ingredient. Legume proteins have appreciable techno-functional properties (e.g., emulsification, foaming, water absorption), which could be affected along with its digestibility during processing. Extraction and isolation of legumes' protein content makes their use more efficient; however, exposure to the conditions of further use (such as temperature and pressure) results in, and significantly increases, changes in the structural, and therefore functional and nutritional, properties. The present review focuses on the quality of legume protein concentrates and their changes under the influence of different physical processing treatments and highlights the effect of processing techniques on the structural, functional, and some of the nutritional, properties of legume proteins.

**Keywords:** legume protein; protein extraction; techno-functional and nutritional properties of protein isolates

## 1. Introduction

With the increase in population and rising awareness of consumption of quality proteinaceous foods, researches for finding sustainable high-quality protein sources are proceeding apace [1–3]. Quality of a protein ingredient is generally described by its ability to meet the requirements of essential amino acids and physiological needs of the organism [4]. However, in this era of climate change, the sustainability features of the proteins have gained even more importance. Hence, alternative proteins such as cultured meat, microbial protein, edible fungi, microalgae, and insect proteins are being explored as new and sustainable sources [2,5]. Although several novel alternatives are available in the current market, plant-based proteins are still being preferred, owing to advantages such as having better consumer acceptability that is related to many subjective reasons, such as taste and health, familiarity, attitudes, food neophobia and social norms [6].

On the other hand, plant sources seem to be more reliable and sustainable protein compared to animal products [7]. Recently, legumes have received much interest owing to their lower cost of production, easy accessibility, health benefits, and their nitrogen-fixing ability in soil [8–11]. They grow globally in diverse agroclimatic zones, from alpine and arctic regions to the equatorial tropics [12]. They can be used as sources of food, flavouring plants, fodder and green manure plants or other industrial raw materials, such as oils, biofuel, timber trees, gums, dyes, and insecticides (Figure 1). Therefore, legumes can be envisioned as super crops, meeting the challenges of climatic changes and the increasing population demands.

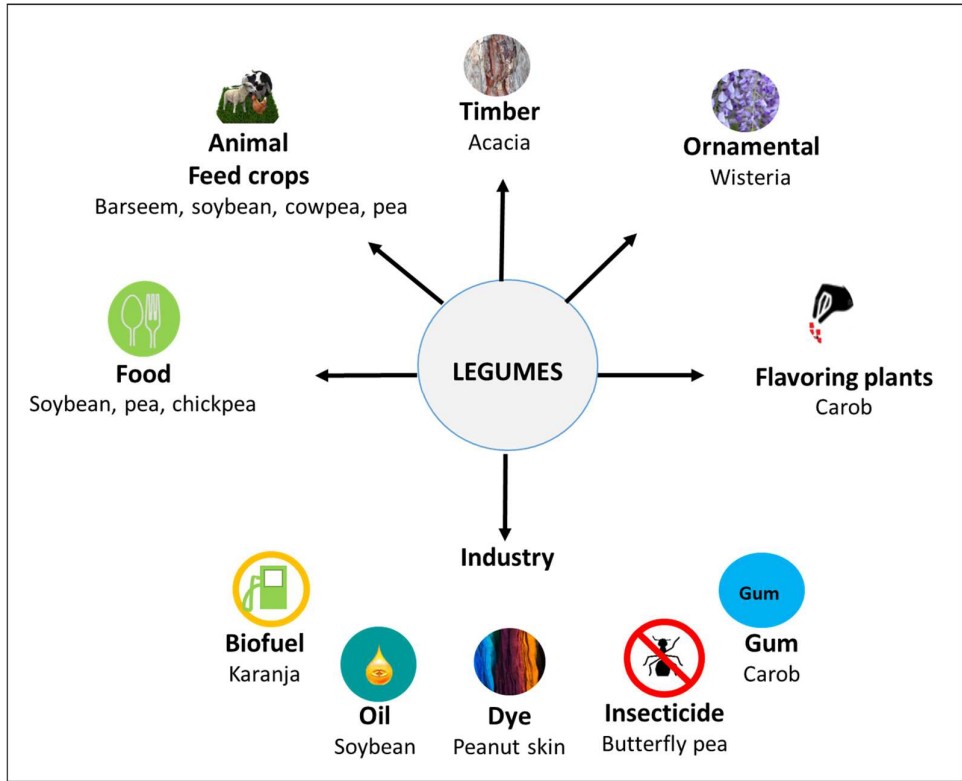

**Figure 1.** Multi-functional utilization of legumes. The scheme was constructed based on [13–19].

Various studies have brought about evidence that indicates the great potential of legumes as a source for providing high quality proteins. Furthermore, this potential highly depends on vital factors such as bioavailability, digestibility, amino acid profile and the possible anti-nutrient content of food/protein extract matrices [20–22]. Raw legumes generally have a low degree of bioavailability, and consequently poor nutritional value [23]. Compared with animal proteins, legume proteins are absorbed at slower rates, which might explain their lower dietary utilisation [24]. Therefore, it is of paramount importance to improve their nutritional quality. The nutritional quality is linked to the chemical composition (including protein/peptide fractions and their proportion; anti-nutrient content, etc.). In particular, from an industrial point of view, the antinutrient is one of the factors that can be influenced by the extraction of protein or by physical, chemical, or biological processes [25–27].

During the extraction processes of legume protein, the structural and functional properties of proteins could be affected due to the intense pH or temperature conditions [25,28]. These events might include physical processing steps such as dehulling, milling, isolation (by air classification, electrostatic separation, hydrocyclone or solvent extraction) and drying (freeze-, spray-, oven- or vacuum-drying) [29–31]. From the industrial perspective, extraction techniques are a part of processing of commercial protein powder [30].

During the manufacturing of protein-containing foods, proteins may be exposed to several processing steps, such as soaking, cooking, baking, drying, pressure-cooking, autoclaving, extrusion and microwaving [23]. The study of changes in the structure and behavior of proteins as a result of the industrially applied physical processes highlights the techno-functional and nutritional parameters that must be carefully considered during product development [32]. Hence, from industrial and consumer viewpoints, understanding protein's behaviour within the food system is absolutely crucial for any food product development.

Therefore, the current review focuses on presenting the effects of extraction, thermal and non-thermal physical processes affecting the structural, functional, and nutritional

characteristics of legume protein(s) isolates, which can give streamlined indications to improve the processing technologies to obtain properly balanced protein-containing foods.

## 2. Legume Proteins

### 2.1. Classification and Composition

Legumes are principally stored in the embryonic cotyledons [33]. The protein content in legumes ranges from 13 to 30 g/100 g dry matter. Some commonly consumed legumes with respective protein contents are pinto beans (15.41%), navy beans (14.98%), black beans (15.24%), cowpea (13.22%), kidney bean (15.35%), chickpea (14.53%), lentils (17.86%), lupin (25.85%) and soybean (28.62%) [34].

Legume proteins can be classified by their sequence, structure, function, conserved motifs, biological activity, solubility, and sedimentation coefficient [35] (Figure 2). According to their biological activities, proteins are primarily of two types: (i) storage proteins, which account for approximately 70% of total seed nitrogen, and (ii) enzymatic, regulatory, and structural proteins, which perform normal cellular activities, including the synthesis of storage proteins [36]. Further, the storage proteins can be classified into seed storage proteins and vegetative storage proteins. The vegetative storage proteins are the proteins that accumulate in vegetative tissues, such as leaves, stems, depending on plant species. It serves as a temporal reservoir of amino acids for use in subsequent phases of growth and development [37]. According to the Osborne classification scheme, legume proteins can be classified based on their solubility and can be fractionated as globulins (70%), albumins (10–20%), glutelins [38] and prolamines [39,40] (Figure 2).

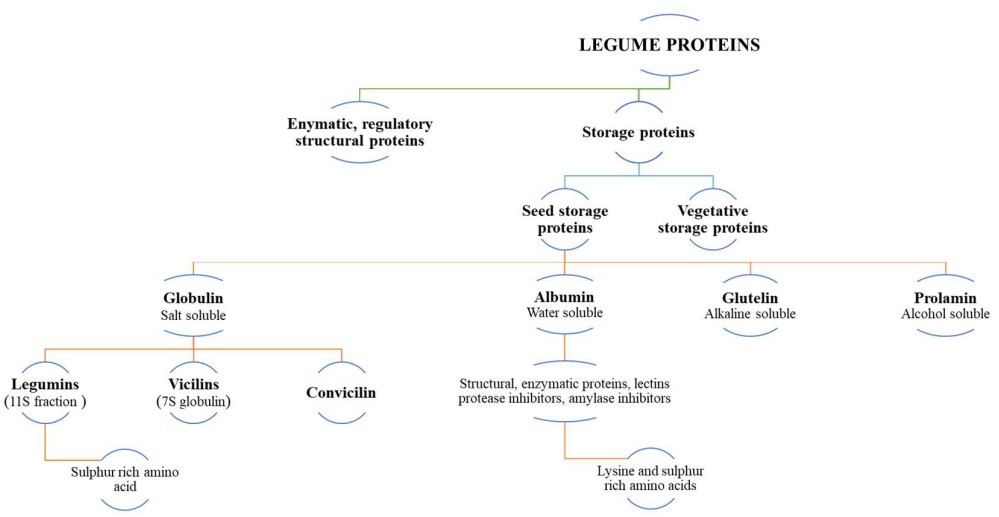

**Figure 2.** Classification of legume proteins [35–37,40–43].

Overall, all legume flours are dominated by albumin and globulin proteins [42,44]. The globulin is the predominant protein fraction in legumes such as soybean, green gram bean, kidney bean and pea [45–47], which can be further classified based on the sedimentation coefficient: legumins (11S), vicilin (7S) and convicilin (Figure 2). Vicilins are found in various legumes, such as kidney bean, mung bean, red bean, cowpea and bambara groundnut, while chickpeas are high in legumins. Pea and faba bean proteins comprise of both vicilins and legumins [47–49].

Since legume seeds contain more globulins than prolamin, they are rich in lysine and tryptophan [44,50] but contain less methionine and cysteine [50]. The nature of the protein depends on their amino acids, which are the basic building blocks of a protein. Amino acids are considered as the indicators of nutritional qualities and functionalities of proteins. It is an important parameter to consider when they are used to improve the nutritional value of a food products [25,51]. There are 20 amino acids, which dictate the quality and nature of protein [52]. Overall, glutamic acid and aspartic acid amino acids are the most abundant

amino acids in legumes. However, tryptophan, methionine and cysteine were found less abundantly. Legumes are a poor source of essential sulphur-containing amino acids [53,54]. Table 1 shows the amino acid composition of different legumes. It is interesting to note that almost all the legumes have similar essential amino acid ratios, as recommended by the FAO [55].

**Table 1.** Amino acid content of various legumes and the recommended amino acid scoring patterns for adults (g/100 g protein) [55–66].

| Amino Acids/Legumes | Chickpea | Soybean | Lentil | Bean | Faba Bean | Lupin | Cowpea | Pea | FAO |
|---|---|---|---|---|---|---|---|---|---|
| Alanine | 4.5–5.2 | 3.615–4.7 | 4.2–4.7 | 3.07–4.89 | 3.97–4.15 | 3.17–3.8 | 4.2–4.67 | 3.54–5.2 | NR * |
| Arginine | 8–9.2 | 6.18–8.3 | 7.6–7.8 | 5.29–6.08 | 8.96–9.46 | 8.51–14.1 | 6.66–7.69 | 4–8.6 | NR |
| Aspartic acid | 10.2–12.1 | 7.13–11.8 | 11.8–13.7 | 9.02–12.94 | 9.28–10.77 | 8.2–15.1 | 10.8–11.44 | 8.06–12.37 | NR |
| Cysteine | 0.4–1.7 | 1.5–2.065 | 0.7–0.9 | 0–0.94 | 0.85–1.33 | 1.12–1.72 | 0.28–0.32 | 0.35–1.8 | NR |
| Glutamic acid | 16.5–17.8 | 9.115–18.2 | 21.4–21.5 | 11.42–17.06 | 15.67–16.51 | 19.19–22.15 | 17.2–18.54 | 8.53–19.7 | NR |
| Glycine | 3.4–4.3 | 3.71–4.4 | 3.6 | 3.19–4.55 | 3.95–4.73 | 3.83–4.77 | 3.8–4.48 | 3.87–5.27 | NR |
| Histidine | 2.7–3.2 | 2.4–3 | 2.2–2.5 | 2.59–3.42 | 2.41–2.61 | 2.31–2.95 | 3.06–3.19 | 1.92–2.94 | 1.6 |
| Isoleucine | 4.1–5.2 | 4.2–5.9 | 3.8–4.1 | 3.42–5.21 | 3.67–3.94 | 2.89–4.62 | 3.75–3.84 | 3.09–4.5 | 3 |
| Leucine | 7.7–9.5 | 7.095–7.9 | 7.8 | 6.72–8.46 | 6.57–7.47 | 5.83–7.3 | 7.65–7.7 | 6.7–7.84 | 6.1 |
| Lysine | 6.7–7.8 | 6–6.58 | 7–7.3 | 4.91–6.48 | 5.97–7.08 | 4.35–4.92 | 5.74–7.5 | 3.41–8.1 | 4.8 |
| Methionine | 0.8–1.6 | 1.1–2.72 | 0.8 | 0.72–1.76 | 0.52–1.06 | 0.35–0.7 | 1.46–2.11 | 0.72–1.6 | NR |
| Phenylanine | 5–6.2 | 3.88–5.8 | 4.5–5 | 4.48–5.91 | 3.98–4.19 | 3.42–4 | 5.75–5.51 | 4.02–5.2 | NR |
| Proline | 3.5–4.4 | 3.63–5.3 | 3.5–4.9 | 2.95–6.49 | 3.86–4.27 | 4.4–5.72 | 4–5.91 | 2.11–5 | NR |
| Serine | 3.3–5.6 | 5.5–6.375 | 3.5–5.2 | 4.59–6.9 | 4.28–4.76 | 4.31–5.98 | 4.5–5.6 | 2.93–5.71 | NR |
| Threonine | 2.7–3.9 | 2.68–4.1 | 3–3.5 | 3.17–4.72 | 2.96–3.4 | 2.9–5.02 | 3.8–4.1 | 1.55–4.46 | 2.5 |
| Tryptophan | 0.6–1.4 | 1–7.64 | 0.7–1.2 | 1.11–1.18 | 0.85–0.87 | 0.49–1 | 0.7–1.11 | 0.61–3 | 0.66 |
| Tyrosine | 2.6–3.1 | 4–4.14 | 3.2–3.3 | 2.75–5.25 | 2.59–2.78 | 3.11–5.1 | 2.92–4.04 | 3.17–3.7 | NR |
| Valine | 3.9–5.2 | 4.4–5.245 | 4.5–5 | 4.58–5.38 | 3.41–4.31 | 2.46–4.2 | 4.68–5.1 | 3.97–5.11 | 4 |

* No recommended score. Underline labelling indicates essential amino acids.

## 2.2. Protein Quality of Legumes

The quality of legume protein is linked to the higher levels of albumin and globulin fractions, owing to the high amount of essential amino acid lysine [45]. The differences in the amino acid composition of these fractions could have an effect on protein digestibility. The studies suggested that globulin has higher digestibility compared to the albumin fraction in lentil and horse gram, which was associated with less cystine content, and therefore less disulfide bonds [67]. In contrast, a negative correlation between globulin content and its in vitro protein digestibility of legumes such as garden pea, grass pea, soybean and lentil was related to their resistance to gastric digestion [68].

However, globulin is hydrolysed into peptides and amino acids during the intestinal phase [68]. For the garden pea, grass pea and lentil, large-size peptides (6–20 kDa) were detected at the end of the gastric phase, but in the intestinal digests, fragments up to 6 kDa were found, except in the case of soybean, where peptides were below 4 kDa [68]. Similarly, cowpea globulins were poorly digested by pepsin, but not by serine proteases [69]. In the case of albumin, a high amount of aromatic (tyrosine, phenylalanine and tryptophan) and β-branched (threonine, valine and isoleucine) amino acids did reduce the digestibility. This was attributed to the restricted enzyme accessibility due to aggregate formation by hydrophobic linkages [67]. Another possible reason for the low digestibility of albumin could be its compact quaternary structure, which is known to impart low digestibility [70]. These studies suggested that the quality of legume proteins could be varied predominantly depending on their amino acid profile, hydrophobic interactions, and the generated protein fractions.

## 2.3. Extraction Techniques

In order to increase the protein content and to reduce the impact of anti-nutrients, the proteins are extracted, including isolates [25] and concentrates [39]. The extraction process depends on many factors, such as pH, temperature, particle size, ionic strength, type of salt used and solvent-to-flour ratio [71]. During the selection of an appropriate

extraction technique for protein, the efficiency of the process (i.e., recovery rate), and the characteristics of the final product (i.e., nutritional quality) should be also considered. For example, the extraction process could involve intense conditions, such as very high/low pH and/or temperature, which could induce damage to the proteins [72]. Additionally, the extraction process must be energy efficient and environmentally friendly [73]. Therefore, all the mentioned factors should be considered before selecting a suitable extraction method.

The extraction can be performed in both dry and wet conditions [71]. Dry extraction consists of milling followed by the fractionation of flour particles into its constituents such as starch, fiber, or protein [74]. Protein enrichment by dry-based extraction technique is typically based on differences in size and density of legume flour [74]. Factors such as starch granule size, uniformity and its interaction with protein molecules (as they dictate the grain hardness) are important parameters affecting its efficacy [75].

For instance, the strong adherence of large amounts of insoluble protein to starch granules might cause difficulty during the separation process [76]. Starch granules in legumes range from 4 to 85 µm [77], whereas specifically for the pulse sub-group it is about 20 µm [78]. Pulses contain a uniform size of starch particles, which makes them suitable raw material for dry fractionation [78]. For pulses, the fractionation of different components can be performed by air classification [71], electrostatic separation, and sieving (sifting) technique [78]. Among these, air classification is the commonly used separation technique. It separates the light fine fraction (protein) from the heavy coarse fraction (starch) [42] (Figure 3).

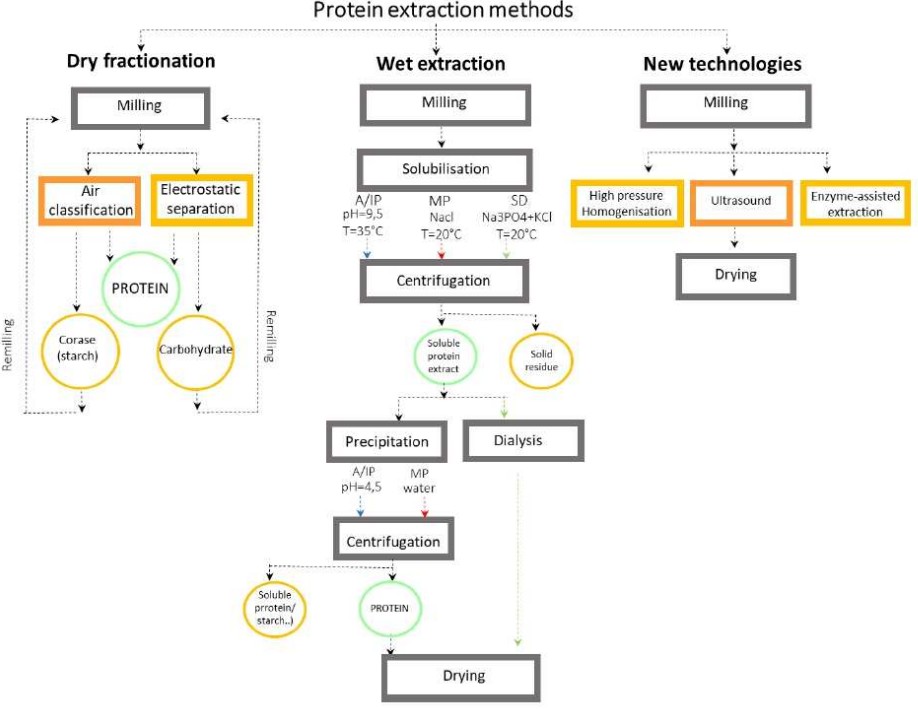

A/IP: Alkaline /Isoelectric precipitation
MP: Micellar precipitation
SD: Salt extraction Dialysis

**Figure 3.** Legume protein extraction methods [29,44,75,79–88].

Seed properties including hardness, particle density, starch granule size, fat content and flour dispersibility influence the protein content during air classification [89]. Therefore, this technique is useful particularly for the production of protein-rich meals and flours (protein content up to about 60%) from non-oilseed legumes like peas [90]. For instance, protein extracted from legumes with low lipid content such as lentil and chickpea exhibited

higher protein recovery rate and yield compared to other legumes such as broad and kidney bean [91]. However, it is difficult to fractionate legumes with high oil content by air classification [82]. In such seeds, oil may promote the cohesion between powder particles through viscous and capillary bridging, which negatively affects the dispersion of individual particles in air [74]. Hence, the air classification technique for lipid-containing legumes might produce protein fractions with less purity [73]. Additionally, oil might soften the grain, giving it a ductile behaviour, and thus increasing milling energy demand [74]. Therefore, dry fractionation of protein is an appropriate technology only for the low lipid legumes. For oil-rich legumes, defatting prior to dry fractionation would be a straightforward approach to obtain higher protein content and purity [74,92].

Furthermore, the difference in the particle size of protein and starch affects the separation process. It was demonstrated that the smaller protein-rich fragments could be separated efficiently [89]. However, the protein separation efficiency reduces by the decrease in size of starch or the increasing ratio of damaged or broken starch granules. As a consequence, their differentiation from protein bodies become difficult [81].

Unlike air classification, the electrostatic separation, which is one of the eco-innovative technologies [93] is based on differences in dielectric properties of flour particles instead of their size and density [94]. However, this difference does not have any advantage on the purity of plant protein isolate/concentrates, including legumes [29]. Based on the methods used to impart surface charge, three main types of electrostatic separators are the induction charging, corona discharge, and tribocharging, based on the methods used to impart surface charge [29].

Another approach for production of protein ingredient is wet extraction (Figure 3). It is an efficient method, which can increase protein content up to 90% and 70% in isolates and concentrates, respectively [71]. This method employs different solvents such as alkaline, acidic, salty aqueous solutions or water as extraction media followed by isoelectric precipitation (IEP) or membrane separation [42,71,95].

In general, to evaluate the separation technique efficiency, the purity and yield of the protein isolate/concentrate should be considered. The protein yield depends on the extraction method and conditions (i.e., particle size of the flour, solubilising agent, pH of solubilisation and precipitation [96], nature of the legume [25,97] and variety [95], classifier wheel speed [98,99], etc.). Many researchers compared the purity of legume proteins that was extracted with different dry, wet and hybrid fractionation methods [29,100]. The purity of protein isolate obtained from dry fractionation is lower than from wet extraction. While higher than 80% protein concentration can be achieved by wet process, dry fractionation results only 54% protein concentration in the case of lupin [73], however, further treatments (e.g., electrostatic separation) can improve efficiency [101]. Dry technique preserves the native state of the protein (i.e., structure and function) [82,89,95] and consumes lower energy and water, and retains the native structure and function of components [78,81,82].

Several modifications can be made to the basic steps of the protein extraction process to enhance its efficiency. For an instance, flowability-aiding agents such as fused silica particles can be added during air classification. Research in [74] showed an increase in the yield of the lupine protein-rich fraction with the addition of flowability aids. Additionally, repetition of the extraction cycle enhanced the separation efficiency as reported by [78]. Researches have showed that the combination of protein extraction technologies could improve its separation efficiency [29]. For instance, the combination of electrostatic separation with air classification could increase the purity of protein isolate compared to air classification alone [92,102]. First, the air classification could remove starch, and subsequent electrostatic separation separated the protein and fiber into oppositely charged fractions [92,102].

However, the combination of dry fractionation with aqueous could be used to obtain fractions with higher protein contents [73,103]. Thus far, several novel techniques, such as direct-assisted extraction, enzymatic-assisted extraction, cavitation-assisted extraction (e.g., ultrasound-assisted extraction, hydrodynamic cavitation extraction), supercritical extraction, pulsed electric field, liquid biphasic flotation, hybrid extraction processes and

microwave-assisted extraction are being explored to increase the efficiency of protein extraction, specifically from food waste [72,104]. A recent study demonstrated that the intervention of ultrasound technology during the extraction of pea protein enhanced its extraction efficiency in short time with low water consumption [105]. Other studies also showed similar outcomes for the extraction yield of soy protein using ultrasonication [85,106], and lupin [107]. The application of the high pressure homogenisation treatment improved extraction yields of soybean protein up to 82%, which was attributed to the reduction in particle size and cell disruption [84].

In addition to the enhance extraction efficiency, the combination of extraction techniques can enhance the functional and nutritional proprieties of the protein isolate. The intervention of ultrasonication during extraction of lupin protein reduced the toxicity and improved its amino acid score [107]. Interestingly, the extraction of pea protein with ultrasound-assisted alkali extraction method improved its functional properties (solubility, water/oil-holding capacity, foaming/emulsifying capacity, stability, and gel formation capacity) with strong antioxidant activities [105]. Furthermore, the application of the alkaline extraction, before the first and the second enzymatic hydrolysis, enhanced the yield of peanut protein extract and their richness in small peptides [83]. Likewise, the extraction of protein from heat-denatured soy meal using protease pre-hydrolysis followed by subcritical water treatment improved its hydrophobic amino acids composition, surface hydrophobicity, interfacial adsorption in addition to an improvement of emulsifying ability and physical stability of emulsion. However, it decreased the protein purity with prolonged hydrolysis [86].

## 3. Processing Affects Structural and Functional Properties of Proteins

During food processing, the variation of conditions (temperature, timing, pressure, pH, etc.) can lead to changes in the structural (Table 2) and consequently the functional properties of proteins (Table 3). In an ideal situation, once the protein is produced in the cytoplasm, its folding will be aided by chaperons and post-translational modifications to develop the structural conformation compatible with its own function. After functional accomplishment, the proteins will become ultimately degraded into amino acids that can be used for the next protein synthesis. This ongoing saga of proteins is regulated by multiple cellular mechanisms at relatively high energetic costs. Extracting proteins from their natural life cycle will render them non-functional and stable, to some extent, so that such a fact stands at the basis of food preservation. The food-specific stability of proteins is paramount to manufacturers but for consumers the importance would fall onto the reutilization of amino acids from proteins. It remains therefore a constant challenge how to match the protein stability and amino acids availability in the case of every foodstuff containing protein extract.

**Table 2.** Effect of processing on the structure of legume protein isolate/concentrate.

| Physical Process | Effect | Protein Extract | References |
|---|---|---|---|
| Heat treatment | Unfolding and denaturation | Cowpea protein isolates | [108] |
| | | Soybean protein isolate | [109] |
| | Denaturation and/or subsequent aggregation Alteration in secondary and tertiary conformational | Kidney, red and mung beans protein isolates | [110] |
| | No dissociation in the protein subunit | Mung bean protein isolate | [30] |
| | Increased α-helix decrease in β-sheets | Soybean Protein Isolate | [111] |

**Table 2.** *Cont.*

| Physical Process | Effect | Protein Extract | References |
|---|---|---|---|
| Extrusion treatment | Denaturation<br>Increase degree of aggregation and crosslinking | Mung bean protein isolate | [112] |
| | Increase in the proportion of β-turn structure<br>Decrease of α-helix and β-sheet | Soy protein concentrate | [113] |
| | | Pea protein isolate | [114] |
| | | Pea protein isolate | [115] |
| | Increase in α-helix<br>Decrease in β-sheet content | Mung bean protein isolate | [112]. |
| High pressure | Unfolding/denaturation/<br>aggregation | Red kidney bean protein isolate | [116] |
| | | Cowpea protein isolates | [108] |
| | | Yellow field pea protein isolate | [117] |
| | Changes in the secondary structure (β-sheets) | Red kidney bean protein isolate | [118] |
| Cold plasma | Compact tertiary structure<br>Higher ordered secondary structure<br>Dissociation of globulins | Grass pea protein isolate | [119] |
| | Oxidation/alteration of the secondary and tertiary structures | Soy protein isolate | [120] |
| | Alteration of the secondary structures | Peanut protein Isolate | [121] |
| Irradiation | Modification of the secondary and tertiary structure | Red kidney bean<br>phytohemagglutinin (lectin) | [122] |
| | Insoluble amorphous aggregates and partially unfolded | Jack bean Concanavalin A | [123] |

**Table 3.** Effect of processing on the functional properties of legume protein isolate/concentrate.

| Properties | Treatment | Effect | References |
|---|---|---|---|
| Water-holding capacity | Heat treatment | Increase | [124] |
| | High hydrostatic pressure | Increase | [108,118] |
| | Cold plasma | Increase | [121,125] |
| | Irradiation (X-ray irradiation) | Decrease | [126] |
| Oil-holding capacity | Heat treatment | Increase | [127] |
| | Cold plasma | Increase | [121,125] |
| Protein solubility | Heat treatment | Increase/Decrease<br>Factors: pH, Temperature | [30,127] |
| | Extrusion | Decrease | [113,114] |
| | High hydrostatic pressure | Increase/Decrease<br>Factor: Protein isolate's nature, Pressure | [116,128,129] |
| | Cold plasma | Increase | [119,121,125] |
| | Irradiation (Gamma/X-ray irradiation) | Decrease | [126] |
| Surface hydrophobicity | Heat treatment | Increase | [130] |
| | Extrusion | Increase/Decrease<br>Factor: Feed moisture, Temperature | [112,113] |
| | High hydrostatic pressure | Increase | [131] |
| Emulsifying activity and stability | Heat treatment | Increase/Decrease<br>Factor: pH | [30,110,124,<br>132] |
| | Extrusion | Stability: Increase | [113,114] |
| | High hydrostatic pressure | Increase/Decrease<br>Factor: Pressure | [116,118,129] |
| | Cold plasma | Increase | [120,121] |
| | Irradiation (Gamma/Electron beam irradiation) | Increase | [126,133] |

**Table 3.** *Cont.*

| Properties | Treatment | Effect | References |
|---|---|---|---|
| Emulsifying activity and stability | Extrusion | Decrease | [114] |
| | High hydrostatic pressure | Increase | [129] |
| | Cold plasma | Increase | [120,121] |
| | Irradiation (X-ray irradiation) | Decrease | [126] |
| Gelling capacity | Heat treatment | Increase | [30] |
| | High hydrostatic pressure | Increase | [108,134,135] |
| | Cold plasma | Increase | [136] |

*3.1. Effect of Thermal Processing*

3.1.1. Effect of Heat Treatment

Thermal processing is one of the most common processing technologies employed in the food industry. It seems to be the most promising processing method, with a great influence on the structure of proteins [137].

Generally, an increase in temperature may lead to rearrangement and partial unfolding of protein molecules [127]. Several studies showed that thermal processing could result in protein aggregation due to disulfide bond formation [108,109,130,132]. This processing method could induce conformational changes in the cowpea, pea and soybean protein isolates when heated at 90 °C, 80–100 °C, and 90 °C and 120 °C, respectively [108,109,132]. The authors suggested that the high temperature could denature the protein isolate as observed in case of soybean protein [109]. Likewise, 95 °C heat processing for 30 min led to extensive denaturation and/or subsequent aggregation accompanied by decrease in free SH and alteration in secondary and tertiary conformational structure in kidney, red and mung beans protein isolates [110].

However, oven drying (50 °C for 24 h) of mung bean protein isolate showed no dissociation in the protein subunit, suggesting its high thermal stability [30]. The secondary structure of the protein mainly consists of α-helix, β-sheets, strands and turns. The α-helix is present in relatively lower proportion [48]. During processing, the amount of each secondary structure present in protein depends on time, temperature, and intensity of heat processing applied [111]. For instance, heat processing of soybean protein isolate increased α-helix and an overall decrease was observed in β-sheets when compared with non-treated counterpart [111]. The secondary structure of protein might also get affected during formulation and long-term storage [127].

Additionally, studies demonstrated that the surface hydrophobicity and hydrophobic interactions increased after thermal processing [109,130]. Researches showed that the β-sheet exhibited a significant inverse correlation with the surface hydrophobicity of soybean protein isolate when heated above 90 °C [111]. Likewise, a negative correlation between solubility and protein denaturation was reported, as well as the high solubility indicating a high ratio of native proteins and low denaturation [124,138].

Since functional properties are directly linked to the structural characteristics of protein, thermal processing might alter the functionality of protein [124]. For instance, the size of soluble aggregates can influence the functional property of the protein. The heat processed soybean protein isolate with a large proportion of higher than 1000 kDa protein aggregates promoted foaming property 2% concentration. However, when the total amount of protein exceeded 60%, those large protein aggregates promoted emulsification. On the contrary, the medium-size aggregates (670 to 1000 kDa) seemed to have a complimentary or supportive role in stabilizing emulsion systems [130].

Heat processing can ameliorate various functionalities such as emulsifying proprieties [124]. Comparing freeze and convective drying at 40 °C and 50 °C, convective drying had higher values of emulsifying activity indices and emulsifying stability indices compared to the freeze-drying extracts [124]. The study conducted on vacuum-dried lentil

protein isolate (at 60 °C and 85 kPa for 48 h) revealed that the processing reduced the solubility with high level of denaturation, which resulted in poor gelling ability when compared to the spray and freeze-dried lentil protein isolate powders. This finding corroborated with the solubility results, since a low concentration of highly soluble protein could make an acceptable gel [138]. Another study revealed that the heat processing of kidney, red and mung beans protein isolates at 95 °C for 30 min improved their solubility and emulsifying activity [110].

For soybean protein isolates, heat treatment using a water bath (90 °C) and autoclave (120 °C) for 15 min denatured the protein isolate, leading to facilitated droplet breakup and decreased droplet size. This treatment was inclined to adsorb on the surface of oil droplets during emulsion storage, which provides a thick and viscoelastic interfacial layer [109]. Another study revealed that heating during spray-drying causes the partial unfolding of globular protein, leading to the exposure of hydrophobic amino acid residues towards the external environment, which increases the adsorption at the oil-water interface and subsequently improved emulsion activity index and stability [30].

Heat processing conditions such as temperature, duration, pH, influence protein structure in different ways, and consequently, their functional properties. For instance, heat pre-treatment (at pH 7.0) enhanced the emulsion-forming ability of isolated pea protein, which led to smaller-sized emulsified oil droplets. On the contrary, pre-treated samples at pH 3.0, 5.0, and 7.0 affected negatively the foaming ability [132].

The difference in thermal stability of proteins depends on species and thermal stability too. For instance, the gelling ability of the globular proteins of pea, soybean and lupin are different, which was due to the variation in their thermal stability. This phenomenon is linked to their structural variations such as amino acid composition, molecular weight, protein subunits and surface hydrophobicity [139]. The difference in the amino acid composition of the protein sub-fraction could reflect in different conformations, which makes it impossible to generalise the type and amount of bonds, secondary, tertiary or quaternary structures present in protein [137]. However, the proportion and the amount of a particular sub-fraction can affect the final functional characteristics. For instance, the preheated mixture of pea vicilin and globulins increased final moduli (elastic (G′) and loss (G′′) modulus) of the acid gels, whereas legumin-enriched samples displayed low gelling properties [140].

### 3.1.2. The Effects of Extrusion Treatment

Extrusion is the most versatile processing technology used in the food industry [141]. It is defined as a high-temperature and short-time process [142]. It is a multi-functional and thermal/mechanical process [143] that enables the control of different processes such as mixing, heating, cooking, shearing, and shaping of products [141]. It can be used to produce pre-cooked and dehydrated foods [144] and has the ability to develop products with better nutritional, functional and sensory characteristics [141]. Several products are made using extrusion technology such as breakfast cereals, savory snacks, crispy flatbread, pre-cooked flours, cereal-based baby food, and textural proteins [141,145]. Currently, many studies are being focussed on the production of legume protein meat extenders and meat analogues using extrusion technology [112,146].

During extrusion treatment, the free sulfhydryl content, particle size, secondary structure, and tertiary structure of protein isolate might change, influencing the functional properties of the legume protein isolate [114,115]. Further, the extrusion could influence the interaction of legume proteins with non-protein matrix, such as microstructure, solubility and emulsifying capacity of mixtures of starch–pea protein isolate [147], soy protein concentrates–rice [148], soy protein isolate–corn [149] and isolated/textured soybean proteins–pectin [150]. These interactions could be affected by the barrel temperature, screw speed, feed rate and feed moisture [146].

The hydrophobic interactions, hydrogen bonds, disulfide bonds and their interactions are the most critical factors that collectively hold the protein structure [114,151]. This

structural assembly within protein is destroyed due to the triple action of high temperature, pressure and shear force, ultimately resulting in protein depolymerisation. These depolymerised proteins recombine under specific temperature and moisture content conditions to form protein aggregates [114].

Among all factors, the feed moisture plays a major role during the extrusion process. For instance, denaturation and formation of mung bean protein aggregates could be controlled by manipulating feed moisture content during extrusion [112]. An extrusion process with low feed moisture content of 30% resulted in extensive protein denaturation, resulting in an increased degree of aggregation and crosslinking [112]. However, similar effects were observed for soy protein concentrate at lower moisture content (18 and 25%) [113]. According to these authors, further increase in feed moisture content to 30%, could cause an increase in the proportion of β-turn structure [113]. For pea protein isolate, the reduction of β-sheet and α-helixes after extrusion under 150 °C and 26% of moisture feed was associated with the changes in the secondary structure [115]. Similar reduction in the β-sheet structure of extruded pea protein isolate was observed at 30% moisture content [114]. The decrease in β-sheet and α-helical structures were related to the increase in protein aggregates [115]. Therefore, it can be interpreted that the changes in the secondary structure during extrusion depend on the protein source as well as on the feed moisture content.

Increasing moisture content could increase the interactions between disulfide bonds with hydrogen bonds and hydrophobic interactions, which reduces the degree of aggregation [142]. The feed moisture content ~49% promoted partial protein unfolding and formation of small aggregates. This partial unfolding could be manifested by increase in α-helix and disulfide bond content and a decrease in β-sheet content [112]. However, studies showed that the high-moisture extrusion could result in the denaturation of protein concentrate [152]. A complete denaturation of mung bean protein isolate was reported along with formation of large protein aggregates when extruded at 60% feed moisture content [112]. A reduction in the α-helices and β-turn was also noticed as this moisture content [114]. Hence, the feed moisture content could directly influence the structural organisation of protein, which might further depend on factors such as temperature.

During the extrusion process, high temperature and shearing causes protein dissociation inside of the equipment; water loss and aggregation after the molten mass exits the die and a sudden drop in temperature. This aggregation usually decreases protein solubility, and hence influences hydrodynamic properties, which are related to the formation of supramolecular assemblies [153]. Studies suggested that the temperature and ingredients during extrusion could affect the pea protein solubility [154]. Extrusion of pea protein isolate at different moisture feed (30, 40, 50, and 60%) and temperature (60 °C, 100 °C, 120 °C, 140 °C, and 175 °C) resulted in protein aggregation, which reduced its solubility [114]. Similar effects were observed for soy protein after processing at 30, 60, 80, 115, and 130 °C with 35% of moisture feed [153]. However, for pea protein isolates, a higher feed moisture content of 55% reduced its solubility [154].

Extrusion cooking at 130–170 °C with 26–35% moisture, reduced solubility of pea protein isolate [115]. Likewise, at 18 and 25% moisture content and 110, 130 and 160 °C temperatures, extrusion cooking decreased the protein solubility of soy protein concentrate compared to the native protein [113]. However, increasing temperature from 110 to 160 °C at 18 and/or 25 % moisture content gradually enhanced the solubility of soy protein [113]. The decrease in the solubility might be related to the formation of heat-induced insoluble aggregates due to the interaction between hydrophobic groups of unfolded protein structures [113]. On the contrary, extrusion of mung bean protein at 144.57 °C with 49.3% feed moisture content improved its solubility [112]. The partial denaturation and formation of small-sized protein aggregates along with increased interactive surface region between protein and water molecules can positively affect the solubilisation of protein, thus improving the overall solubility [112]. Therefore, it can be concluded that the solubility of protein isolate depends on the moisture feed and temperature, which can influence the aggregate size or nature, and ultimately the bioavailability of amino acids from proteins.

The structural modifications during the extrusion process can affect the protein solubility and hence its hydrodynamic properties such as water absorption capacity, emulsification and foaming properties. For example, a higher water absorption capacity of isolated soy protein and mung bean protein was attributed to the higher amount of protein solubility [155]. Similarly, the protein denaturation of the extruded faba bean concentrate (high extrusion moisture content) was translated by a high-water binding capacity [152]. In addition to the solubility, the hydrophilic–hydrophobic amino acids and polar/non-polar amino acids can serve as a useful indicator of the functionality of protein, such as water and fat absorption, and surfactant properties [30,155]. The study showed that the temperature at the last heating zone of extrusion equipment and moisture content significantly affected the water binding capacity of the meat analogues, which was in negative relation with temperature (120 °C to 140 °C) [152].

The foaming properties of a protein can be characterized by measuring the foaming capacity and foaming stability [156]. Foaming capacity is affected by the solubility of the protein isolate [114]. Compared with native pea protein, the foaming capacity of extrudate of pea protein hydrated at 30, 40, 50 and 60% moisture content reduced due to the decrease in its solubility. However, the extrudate with 30% moisture content demonstrated the highest foaming stability [114]. This finding is correlated to the irreversible thermal denaturing of proteins, which might result in the unfolding of the peptide chain. The unfolding phenomenon led to the structural rearrangement within the machine barrel to form protein aggregates. These aggregates are adsorbed on the air–water interface to create a network structure, thus improving the stability of the foam. Additionally, the insoluble protein particles play a favourable role in the stability of the foam [114].

Moisture content seems to have a lubricant role during the extrusion process. At 30 and 40% moisture content, protein is denatured by raising the temperature from 60 to 175 °C, which exposes hidden hydrophobic groups, thus increasing the emulsifying activity index and emulsifying stability index [114]. In contrast, the increase in the extrusion temperature from 110 to 130 °C decreases the emulsifying stability of extruded soy protein emulsion, which could be related to the formation of larger particles when the extrusion temperature increases to 130 °C [113].

Compared with soy protein concentrate extruded at 25% of feed moisture, soy protein concentrate extruded at 18% feed formed unstable emulsion that can be related to the particle droplet size in emulsions. The emulsions fabricated by extruded soy protein concentrate at 25% feed moisture had small droplets size which promotes more stability against coalescence and flocculation [113]. Similarly, the droplet size of the emulsions produced by soy protein was reduced after extrusion treatment at 30, 60, 80, 115, and 130 °C and 35% of moisture feed [153].

The surface hydrophobicity is correlated to functional properties of protein such as emulsification and foaming [112]. It is a physicochemical property that greatly determines the tendency of protein molecules to aggregate and thus to lose solubility [157]. The soy protein concentrate extruded at 110, 130 and 160 °C with 18 and 25% moisture contents exhibiting lower surface hydrophobicity compared with the untreated one. This was attributed to the protein–protein interaction inside the extruder and the formation of protein aggregates, which hide the hydrophobic sites on the surface of protein and inhibit its accessibility [113]. In contrast, a slight increase in the soy protein concentrate's superficial hydrophobicity was reported after extrusion with single-screw extruder at 30, 60, 80, 115, and 130 °C and 35% moisture [153]. For mung bean proteins, the increase in feed moisture content during extrusion effectively increased surface hydrophobicity by denaturing them and exposing more hydrophobic domains, consequently making the protein structure more flexible [112]. Moreover, the increase in the moisture content from 18 to 25% during extrusion-enhanced surface hydrophobicity, which was related to the formation of larger protein aggregates [113].

*3.2. Effect of Non-Thermal Technologies*

3.2.1. Effect of High-Pressure Treatment

High-pressure processing (HPP) is an interesting alternative to traditional processing [158]. It is a non-thermal technology that has several applications including the modification of proteins structure [159]. High hydrostatic pressure (HHP), high-pressure homogenization (HPH) and ultra-high-pressure homogenization are improved forms of HPP [160].

HPP induces gradual unfolding and denaturation in the structure of protein molecules [108,116]. This effect was observed when the yellow field pea protein isolate was treated under 600 MPa [117]. Interestingly, HPP at similar pressure (400 and 600 MPa) completely denatured soybean protein isolates [131]. Indeed, for the same protein isolate a greater extent of protein denaturation, aggregation, and network formation occurred with increasing pressure level (250–550 MPa for 15 min), due to protein tertiary and quaternary conformation changes. Pressure treatments above 350 MPa induced protein denaturation and subsequent gel structure formation [135]. Another study conducted on red kidney bean protein isolate demonstrated that the HPP had little effect on its structure when treated below 600 MPa. However, the increase in pressure up to 600 MPa led to the aggregation of protein molecules/monomers [118]. Likewise, HPH could cause unfolding of lentil proteins at 50–150 MPa [129]. On the contrary, a study conducted on lentil protein isolate showed that HPP could not significantly alter protein structure, even when exposed at high level of 600 MPa for 15 min. The authors suggested that HPP might not have affected covalent bonds, and thus, the primary protein structure did not alter after the processing [161].

Since HPP can alter protein structure, it can induce several subsequent changes in its physico-chemical properties, such as hydration, hydrophobicity, and hydrophilicity [159]. Some of the researches employ solubility and surface hydrophobicity as the indices of degree of denaturation caused by high pressure; [131,162] reported that HPP (200–600 MPa at pH 8) increased surface hydrophobicity and protein aggregation accompanied with a decrease in free SH and partial unfolding of 7S and 11S fractions of soybean protein isolate. Interestingly, HPP at pH 3 resulted in partial denaturation of protein, leading to decreased thermal stability, and increase in protein solubility [131].

Protein solubility is a critical factor in the hydrolytic yield of proteins that may get affected by HPP. HPP of lentil protein concentrates at 100, 200 and 300 MPa exhibited similar soluble protein content as control. However, increase in pressure (400 and 500 MPa) significantly reduced the soluble protein content, probably due to the formation of insoluble protein aggregates [128]. On the contrary, the solubility of red kidney bean protein isolate was significantly improved at 400 MPa or higher, possibly due to formation of soluble aggregate from insoluble precipitate [116]. A study conducted on lentil protein isolate demonstrated that the HPH at 100 MPa could increase its solubility as a consequence of protein unfolding and increased solvent–protein interactions. However, HPH at 150 MPa significantly decreased the solubility, probably as a consequence of the over-processing effect of pressure [129].

In the case of water-holding capacity and gelation property, HHP (200, 400 and 600 MPa) was found to be more efficient than thermal processing in cowpea protein [108]. HHP led to the generation of compact aggregates at high protein concentrations (10.5–12.0 g/100 g), which would have increased the ability of protein to interact through hydrophobic interactions, ultimately producing more elastic gels [134]. HHP aids the production of gels at lower temperatures, which make cowpea proteins useful to texturize hot-serving foodstuff [134].

Emulsification property is another functional property of the proteins which might be affected by HPP [116]. Emulsification is generally known as dispersing one liquid in another immiscible liquid [163]. Factors such as surface hydrophobicity, partial denaturation and disordered structure can affect the potential of protein for the adsorption at the oil–water interface [164]. HPP of kidney bean protein isolate at 200 and 400 MPa significantly increased its emulsifying activity and stability index [116]. Another study showed similar effects of HPP on soybean protein, particularly on β-7S and A-11S polypeptides. However, the authors suggested that the alterations in emulsification properties could be

pH dependent [164]. Since the pH determines the net protein charge, especially below and above the isoelectric point, it might influence the adsorption phenomenon [117]. The HPP at neutral pH could improve the emulsifying activity of soy proteins isolate [162]. However, HHP of yellow field pea protein isolate at pH 3.0 improved emulsion quality with oil droplet size of 26–68 μm, along with higher foaming capacity [117]. Apart from pH, intensity of pressure can also influence the functional properties of legume proteins. A study conducted on red kidney bean protein isolate revealed that the functional properties such as water-holding capacity, and emulsifying increased with the increase in pressure intensity from 200 to 600 MPa, unlike the foaming properties that decreased [118].

### 3.2.2. Cold Plasma

Generally, cold plasma is being employed for improving antimicrobial activity, structural alterations, surface decontamination, etc. At present, the application of cold plasma is being expanded by combining it with other technologies such as nanotechnology, pulsed electric field, pulsed light and ultrasound [165]. Cold plasma is a non-thermal processing technology [136,166], which utilizes energetic and reactive gases [167]. In this technology, a simple gas (such as air or nitrogen) may be used, or the system may rely on a mixture of noble gases, such as helium, argon, or neon [168]. Through ionization of inducer gas with electrical discharges, cold plasma consisting of electrons, ions, neutral particles, and free radicals can be obtained [120].

Studies demonstrated that cold plasma could change the secondary and tertiary structure of grass pea protein isolate, imparting more thermodynamically stable structure. This was attributed it to the increase in hydrogen bonding in the α-helix and β-sheet structures [119]. However, another study reported that the atmospheric cold plasma induced reactive oxygen species-mediated oxidation of soy protein isolate, which ultimately altered its secondary and tertiary structures [120]. For peanut protein isolate, the primary structures were unaltered under cold plasma treatment, unlike the secondary structure [121]. The modification of the secondary structure manifested in an increase in the content of β-sheets and random coil and a decrease in the content of α-helix and β-turn [121].

Atmospheric pressure plasma could improve the functional properties of pea protein in term of solubility [125]. However, the combination of the thermal and plasma-induced pH conditions lead to an increase in protein solubility to 327% [125]. Concomitant to the previous mentioned study, cold plasma processing of grass pea protein isolate for 60 s, increased the protein solubility. It was ascribed to the reduction in protein particle size and the increase in protein surface charge [119]. Similarly, the atmosphere cold plasma treatment improved the solubility of peanut protein isolate [121,169]. The maximum of solubility was reached after 7 min of treatment, with a 12.17% increase comparing with untreated samples [169].

Additionally, the unfolding of protein structure, exposure of active sites, and binding of water micelles to protein molecules could improve the water-holding capacity and emulsion stability of the protein [121]. Another study conducted by [136] showed an improvement in gelling capacity of cold plasma-treated pea protein. Their finding indicated that native pea protein concentrate could not form gel at 90 °C, while the cold plasma could induce a better gelling property. The gels from the processed sample exhibited homogeneous three-dimensional network structure with interconnected macropores. However, gels prepared from processed samples at 80 and 90 °C possessed good mechanical strength and viscoelasticity along with high water-holding capacity [136].

Another study showed that the cold plasma processing could improve the interfacial and emulsifying properties of grass pea protein isolate in terms of thermodynamic stability of protein on interface and globulin dissociation. It also increased oil-droplet surface electrical charge [119]. For peanut protein isolate, the improvement of the emulsifying properties was related to the increased random coils [121]. The intensity of changes in the functional properties of protein seems to be time dependent. For instance, cold plasma processing of soybean protein isolate for 1 min improved its emulsifying and foaming

properties. However, prolonged processing for 5 and 10 min led to the subsequent reduction in both interfacial characteristics [120].

### 3.2.3. Irradiation

Irradiation is a highly efficient and low cost technology with the advantages of short processing time [105]. Gamma irradiation, X-rays and electron beam are the most commonly used radiation treatments in food industries. The term 'food irradiation' is used to describe a process where food is exposed to ionizing energy, just utilizing gamma photons, machine-generated X-rays or accelerated electrons [170]. In case of γ-irradiation, the rays are emitted from the radioisotopes 60Co and 137Cs, or high energy electrons; however, X-rays are produced by machine sources [171]. Depending on the absorbed radiation dose, irradiation can reduce the storage losses, extend the product shelf life and improve microbiological and parasitological safety of foods [171]. Despite being declared as safe and nutritionally adequate by numerous international expert groups, the commercial use of irradiation is still limited [170]. Therefore, there is limited literature available on the effect of γ-radiation on the properties of protein, specifically legume proteins.

The treatment of soy protein isolate by γ-irradiation caused the disruption of the ordered structure and induced degradation, cross-linking, and aggregation of the polypeptide chains [172]. Likewise, a high dose of γ-irradiation of jack bean Concanavalin A (lectin) reduces its β-rich secondary structures. This indicates an accumulation of completely unfolded and fragmented peptides that have a tendency to form insoluble amorphous aggregates [123]. Similarly, γ-irradiation of red kidney bean phytohemagglutinin resulted in complete destruction of secondary and tertiary structure at a dose superior or equal to 10 kGy [122].

Furthermore, studies showed that irradiation treatments, such as γ-irradiation, X-rays and electron beam, could influence the functional properties of legume proteins. The treatment of soy proteins concentrates with γ- (1 kGy intensity) and X-ray irradiation for 15 min enhanced the emulsification activity and foaming capacity; however, it decreased the protein solubility [126]. Similarly, the electron beam irradiation treatment of pea protein hydrolysates increased its emulsifying properties with a maximum effect at 50 kGy. Additionally, both foaming capacity and stability increased with the increase in irradiation dose [133]. A reduction in the water-binding capacity of the gamma-irradiated soy protein concentrate was experienced, which was attributed to the modification in the secondary and tertiary structures [126].

## 4. Effect of Extraction and Processing on the Nutritional Characteristics

The nutritional value of plant-based food depends on the amount of proteins, the specific distribution of the amino acids, and, most importantly, the bioavailability of amino acids [20,23]. The latter is determined by digestibility, that is measured to assess the susceptibility of protein to proteolysis. Therefore, highly digestible proteins can provide more amounts of amino acids for absorption due to their higher degree of hydrolysis, thereby providing better nutritional value [173].

In general, legume proteins have low in-situ digestibility, which could be related to both external and internal factors [174]. The physical inaccessibility due to the entrapment in, for example, intact cell structures, the presence of antinutritional factors [174] and the protein interactions with other compounds such as carbohydrates, lipids and especially anti-nutritional factors (anti-nutritional proteins (trypsin inhibitors, lectins, etc.) and the anti-nutritional chemicals (tannins, phytates, and polyphenols)) are among the external factors [23]. These factors could be influenced by the extraction and/or during other processing steps of legumes grains (Figure 4).

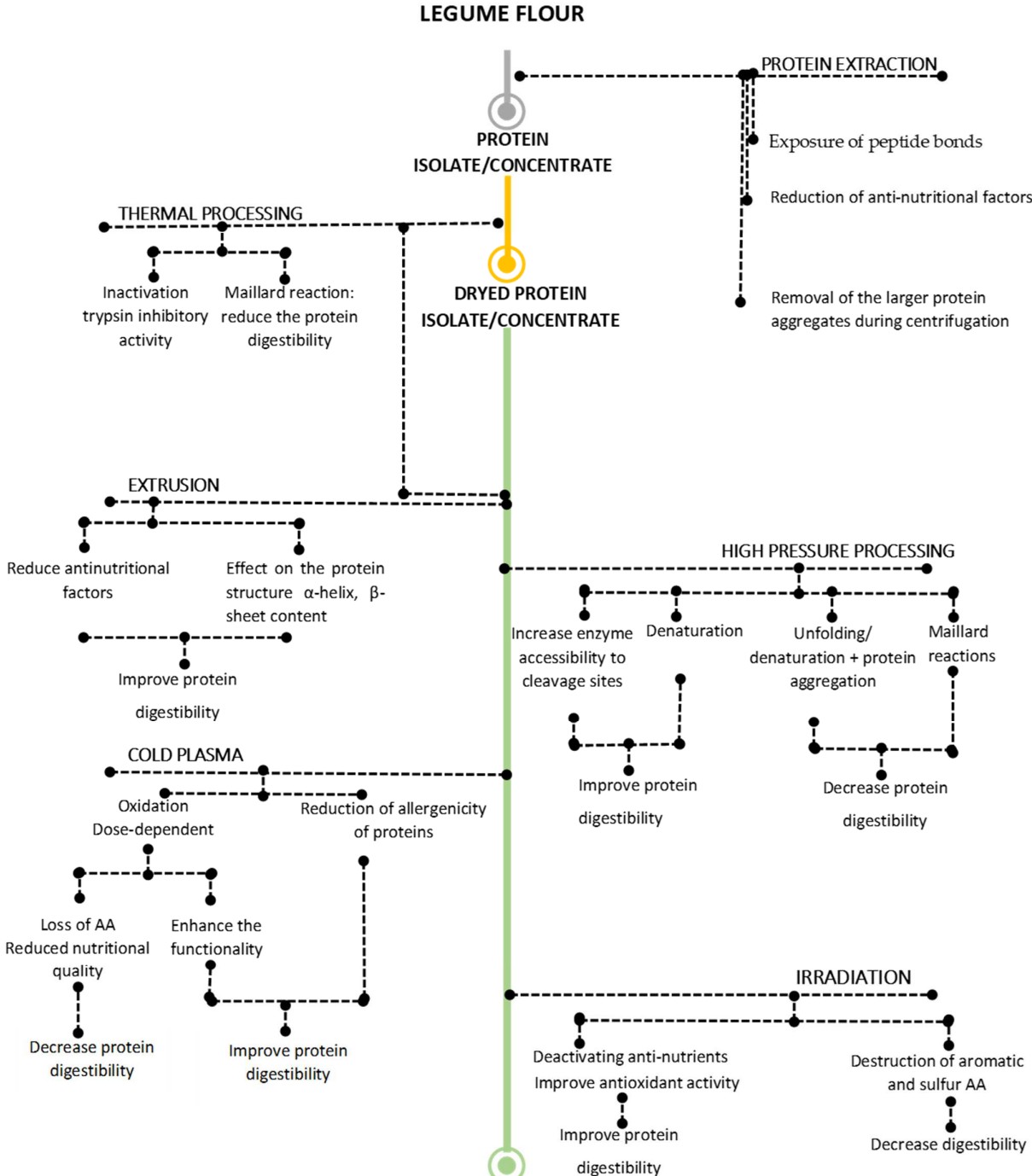

**Figure 4.** Effect of extraction and processing techniques on the digestibility of the protein isolate/concentrate [20,21,30,49,116,120,133,137,161,174–182].

### 4.1. Effect of Extraction

The extraction methods, can influence the in vitro protein digestibility [22]. The changes in the protein digestibility could be related to the modifications in the structure during processing. The literature suggests an increase in protein digestibility after their extraction, which is related to the changes in exposure of peptide bonds and increase in accessibility to susceptible protein sites [182]. The higher digestibility of the lentil protein isolate in comparison with lentil flour might be attributed to variations in the processing such as thermal treatment and spray drying for isolate preparation [182]. However, the authors emphasized that this difference in digestibility might largely depends on the re-

duction of anti-nutritional factors [182]. Hence, the extraction of protein can significantly influence its digestibility. For instance, protein extracted from African mesquite bean using isoelectric precipitation exhibited low digestibility compared to micellization precipitation. This was suggested to be the consequence of the high pH achieved by sodium hydroxide, which resulted into amino acid cross-linking, and subsequent formation of lysinoalanine from lysine and dehydroalanine via serine and cysteine degradation [22]. Furthermore, approaches including digestibility model systems and quantification methods, can offer further details on the digestibility of protein isolates [182].

Additionally, protein extraction has been found to reduce the content of anti-nutritional factors that might subsequently lead to improved digestibility [22,183]. The isolation techniques such as alkaline extraction, iso-electric precipitation and ultrafiltration seemed to lower trypsin inhibitor in lentil protein concentrates and isolates [182,183]. Moreover, the size of the protein bodies seems to be associated with its digestibility. The study showed that the protein extracted from lentils using isoelectric precipitation had a lower amount of larger molar mass proteins (>699 kDa), compared with the dry-milled flour, that is often less digestible and may be often associated with intolerance and/or allergy [184].

### 4.2. Effect of Processing

4.2.1. Effect of Thermal Processing

Heat Treatment

Proteins vary in proteolytic susceptibility and hence their digestibility, owing to the differences in their origin, nature and structure [23]. The main structural component of legume protein is β-sheets and the literature showed its strong inverse correlation with protein digestibility [185]. The disappearance of intramolecular β-sheets could lead to the increase in protein digestibility since it influences the accessibility of enzymes for digestion [137]. Further, the unfolding of the globular protein subunits could enhance its digestibility as it might expose the otherwise buried hydrophobic groups to pepsin [186]. A study conducted on vicilin-rich protein isolated from *Phaseolus* legumes (kidney bean, red bean and mung bean) showed that the thermal treatment improved their nutritional property, which was largely dependent on their amino acid composition and protein aggregation. The heat-induced protein aggregation varied with the type of protein isolate, and the formation of disulphide bonds.

The nutritional relevance of the legume protein such as the protein-specific amino acid content could also be influenced by the thermal processing [49,187]. The roasting of bambara groundnut protein isolate reduced its protein content, which was attributed to the protein aggregation. The larger protein aggregates can be removed during centrifugation before isoelectric precipitation step [49]. The thermal processing seemed to improve the nutritional properties of *Phaseolus* legumes and promoted protein aggregation [187]. Another important factor affecting the nutritional property would be the anti-nutritional factors, which could be reduced by using thermal processing [21]. The thermal treatment of commercial soybean protein products could inactivate up to 80% of trypsin inhibitory activity [21]. However, in case of lentil and faba bean protein concentrates, the thermal treatment at 95 °C for 15 min could led to reduction in trypsin inhibitor activity up to 86% and 78%, respectively, when compared with untreated samples [178]. Interestingly, there is a possibility that thermal and/or alkaline processing led to the formation of compounds such as Maillard reaction compounds, D-amino acids and/or lysinoalanine, which reduced the protein digestibility by up to 28% in rats and pigs [21].

Extrusion Treatment

The primary reason for the improvement in digestibility upon extrusion has been linked to the inactivation of anti-nutritional compounds and to the thermal denaturation of proteins due to high temperatures and shearing [180,188]. The reduction in anti-nutritional factors was noticed during the processing of protein concentrates and extruded products [180,181].

The extrusion process significantly improves the in vitro protein digestibility of lupin protein extrudates compared to that of raw material mixture [180]. Similarly, the extrusion cooking of soybean protein concentrate significantly improved the in vitro protein and starch digestibilities and significantly reduced some anti-nutritional factors of the formulated meals [181]. Partial denaturation of mung bean protein isolate improved its digestibility due to the increased α-helix along with reduced β-sheet content, while stronger gel-forming behaviour was also experienced due to the increased disulfide bond content, frequency-independent viscoelasticity and shear thinning effect [30]. However, another study on the mice demonstrated that the extrusion process did not decrease the protein digestibility of soy protein isolate, and hence could not affect the growth rate of male and female mice [189].

Barrel temperature and feed moisture are two other factors, which can affect the protein digestibility during the extrusion process. The increase in barrel temperature was found to decrease the in vitro protein digestibility of lupin significantly, which could be due to non-enzymatic browning reactions and thermal crosslinking [180]. On the other hand, feed moisture content is an essential factor that should be adjusted during the extrusion of protein ingredients to achieve desirable quality and product performance as a vegetarian-based meat extender [30]. In particular, increasing water feed from 40% to 55% did not show significant difference in in vitro protein digestibility, whereas the sample with 68% water feed exhibited significantly higher in vitro protein digestibility. At higher water feeds (~60%), protein aggregation could have been reduced, which enhanced protein digestibility [180].

Concerning the amino acid composition, the amounts of sulfur-containing amino acids, including cystine and methionine of plant proteins such as soybean and pea tend to increase after extrusion, while the amounts of acid-stable amino acids of texturized vegetable proteins tend to decrease [155]. In contrast, [154] found that the extrusion had no effect on the degree of hydrolysis and amino acid composition indicating that the thermal and mechanical energy during extrusion did not cause the formation of peptide bonds or the degradation of amino acids due to the Maillard reactions.

### 4.2.2. Effect of Non-Thermal Processing
### The Effect of High Hydrostatic Pressure Processing

High hydrostatic pressure (HHP) involves absorption of water and induction of heat within the protein, which result in changes in its conformation, thereby impairing its susceptibility to proteolysis [190]. The changes in protein structure during HHP, such as denaturation and formation of random coil, can have a positive effect on protein digestibility [137]. However, the aggregation and Maillard reactions during HHP can lead to a decrease in digestibility [137,174].

The pressure-induced gels (600 MPa/5 °C/4 min) exhibited more accessibility to gastric proteolysis than heat-induced gels [177]. For lentil protein, pressure treatment (300 MPa for 15 min) increased the enzyme accessibility to cleavage sites of protein, which improved the degree of hydrolysis and their antioxidant activity [161]. In contrast, pressure treatment of red kidney bean protein isolate with 200 MPa or above significantly decreased its digestibility. The decrease in digestibility was due to the protein unfolding or denaturation and protein aggregation but it was noticed only when the pressure was increased above 200 MPa and the incubation time was long (120 min) [116].

### Cold Plasma

The digestibility and bioavailability of the amino acids is related to the nutritional properties of the protein since their degradation; fragmentation or cross-linking reactions can influence the digestibility of proteins [176]. Cold plasma treatment may trigger changes in the amino acid side chains, such as the –SH groups [119,121]. In general, these changes are manifested by a reduction in the free –SH groups due to oxidation of proteins [120]. The aromatic and sulfur-containing amino acid side chains are known to be particularly

susceptible to oxidation, given the extreme sensitivity of cysteine residues [119,121,191]. Similar reduction in cysteine was reported in grass pea protein isolate, probably due to the oxidative effect of ozone produced during plasma discharge [119]. Plasma treatment of pea protein isolate resulted in alteration in fluorescence spectra of tryptophan, which was attributed to the oxidation and the fluorescence quenching [125]. Similarly, cold plasma treatment of pea protein concentrate isolate decreased its fluorescence intensity, which was related to tryptophan oxidation due to the elimination of hydrogen atom on its indole ring by radicals [136]. The cold plasma induced-oxidation of sulfur-containing amino acid residues, such as cysteine and methionine resulted in the degradation of sulfhydryl groups [120].

During cold plasma treatment, the oxidation induced loss of amino acids that can lead to a reduced nutritional quality of protein [176]. However, the effect of the oxidation is dose-dependent [179]. Mild oxidation might enhance the functionality and might not have a significant impact on the nutritional quality of proteinaceous foods since the extent of the loss of essential amino acids is limited [179]. Several studies reported the reduction of allergenicity of proteins after the application of non-thermal treatments due to the changes in the antibody's binding ability [192]. Among these treatments, the application of cold plasma was successful in inactivating allergens in most proteins [176]. For soybean protein isolate, the application of atmospheric cold plasma at 120 Hz for 5 min decreased the level of IgE-binding by up to 75% compared to the control [120]. Another study stated that the cold atmospheric pressure plasma was the most effective in the reduction of Gly m5 immunoreactivity comparing with pulsed ultraviolet light and gamma-irradiation (3–100 kGy) [175].

Irradiation

Similar to the functional properties, the effect of irradiation on nutritional aspect of protein concentrates is less studied area and limited to $\gamma$-irradiation and electron beam. The literature suggested that irradiation levels up to 10 kGy could be an effective measure in deactivating anti-nutrients such as protease inhibitors, lectin, phytic acid, non-starch polysaccharides and oligosaccharides without altering the nutritional quality of the food [178]. Higher than 25 kGy, $\gamma$-irradiation treatment of soy protein isolate reduced its allergenic proteins [175]. Moreover, 30–50 kGy irradiation reduced the potential of red kidney bean lectin to agglutinate rabbit blood erythrocytes [122]. However, [20] reported that the $\gamma$-irradiation negatively affected protein digestibility, which could be due to the destruction of aromatic and sulfur containing amino acids. On the other hand, the electron beam irradiation influences protein structure, which could be manifested in the form of improved antioxidant activity pea protein hydrolysates [133]. Despite these positive changes of irradiation, the legislation has allowed 10 kGy as maximum dose limit of $\gamma$-irradiation for specific commercial food items [175].

## 5. Conclusions

Due to the rapidly growing numbers of the world population, the consumer-specific health awareness, the scientific knowledge advancement in the field of nutrition and improvement of the food manufacturing concepts and technologies, it clearly indicates that the proteins are once again back in the forefront of research. On a larger scale, we also have to address the global warming-induced changes to crop production and the declining health conditions of populations. Moreover, there are other contrasting and even moral issues to consider related to the relevance of proteins in the food chain, including those of food waste, the declining popularity of animal foodstuffs and the increasing relevance of a plant-based diet, all of which is more specific to developed countries. Conversely, in developing countries, food security and safety emerge as the major restrictive factors that must be faced and through multiple solutions ensure the sustainability of social and economic progress. We also have to admit that the focus on diets advocating for proteins being the most critical macronutrients, is shifting towards the bioavailability of essential

and non-essential amino acids [193]. The outmost significance of some amino acids for vital functions is obtaining increasing scientific evidences. For example, amino acids such as glutamine and glutamate were proposed to interfere with the onset of autism spectrum disorder [194], but the glutamate can also regulate the glutamylation of microtubules affecting cytoskeletal functions and cancer metastasis [195].

Legumes are one of the most important sustainable sources of protein, but, by any means, attention should be paid to their digestibility, knowing that several endogenous and exogenous factors have negative influences. The prime examples of these factors are the interaction of proteins with starch and lipids or the damaging effect of the harsh extraction conditions (pH and temperature). Nevertheless, physical food processing techniques, such as wet or dry heat treatment, irradiation or high-pressure treatment, could produce protein extract matrices that would confer additional nutritional benefits regarding the delivery of amino acids, dietary fibers and polyphenols.

**Author Contributions:** Conceptualization, P.S. and C.N.; writing—original draft preparation, C.N. and J.S.; writing—review and editing, P.S., M.H.K. and E.M.; supervision, P.S. All authors have read and agreed to the published version of the manuscript.

**Funding:** This research received no external funding.

**Data Availability Statement:** Not applicable.

**Conflicts of Interest:** The authors declare no conflict of interest.

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
