# Peer review of "Legume Protein Extracts: The Relevance of Physical Processing in the Context of Structural, Techno-Functional and Nutritional Aspects of Food Development"

_processes, doi:10.3390/pr10122586_

Round 1
Reviewer 1 Report
This manuscript described an interesting and meaningful topic. However, a large part of the main document was devoted to the basic information and extraction techniques of legume proteins, rather than focusing on the physical processing methods and the impacts of physical processing on the protein properties. There should be a more explicit and systematic literature review of physical processing, and the effects of different methods on structural, nutritional and techno-functional properties should be discussed separately.
Author Response
We would like to thank the reviewer for the careful reading of this manuscript and for the constructive suggestions which helped to improve the quality of this manuscript.
As in the assessment of the protein content of foods should not only from quantitative, but much more qualitative aspects be considered, it is important to dedicate a short part (one page) of the review to present the composition of legume proteins, the main ways of their extraction and their effects on the nutritional and technofunctional quality. The extraction method, its conditions and conservation technology (i.e. freeze drying, oven) of the extracted protein can also make them react differently during food processing. The changes in nutritional value and technofunctional behaviour is strongly correlate to the structural changes, therefore the structural properties of native legume proteins are important to summarize before the evaluation of their changes during post processing and it is hard to discuss them together. This correlation is the reason that why we chosen the joint discussion of structural and functional changes of protein of concentrates and isolates under different processing circumstances. To the separate presentation of structural changes of these protein products we added a summarizing table about the effect of physical treatment on the protein structure too (table 2).
In the paper our main aim was to summarize of effect physical processing methods on the technological and nutritional value of legume protein extract and isolates and chapter 3 and 4 are dedicated to the in-depth assessment of connections between protein properties and physical circumstances of treatments (temperature, pressure, radiation, etc.). We try to make this more specific by modifying the title and abstract.
Please, see the attached modified manuscript.

Reviewer 2 Report
effect of different processing techniques on % of protein and its quality should be more clear and in a special section
Author Response
We would like to thank the reviewer for the careful reading of this manuscript and for the constructive suggestions which helped to improve the quality of this manuscript.
This review paper does not aim to compare the different kinds of protein extraction techniques applied for legumes, but to assess the effect of further processing of legume protein concentrates and isolates, how the different physical food processing techniques and conditions (T, pH, pressure etc.) influences the structural, functional, and nutritional properties of protein. Therefore, in all the selected research papers, the previously extracted protein (protein isolate/concentrate) was treated and not the legume flour. The further processing no not influence the protein content, but the composition what is discussed detailed in chapter 3. The impact of the extraction techniques on the yield and purity of the protein extract is assessed briefly in the chapter 2.3 and the summary about the benefits and disadvantages of dry and wet methods has been completed. We also made modifications in the title and the abstract to make the aim of the review paper clearer.
Please, see the attached modified manuscript.

Round 2
Reviewer 1 Report
The revision has been improved. It could be accepted after minor fromat checking.